# TractoTransformer: Diffusion MRI Streamline Tractography using CNN and Transformer Networks

Itzik Waizman[1]    Yakov Gusakov[1]    Itay Benou[1]    Tammy Riklin Raviv[1]

[1]The School of Electrical and Computer Engineering, Ben-Gurion University of the Negev
{itzikwei, gusakovy, benoui}@post.bgu.ac.il, rrtammy@bgu.ac.il

## Abstract

White matter tractography is an advanced neuroimaging technique that reconstructs the 3D white matter pathways of the brain from diffusion MRI data. It can be framed as a pathfinding problem aiming to infer neural fiber trajectories from noisy and ambiguous measurements, facing challenges such as crossing, merging, and fanning white-matter configurations. In this paper, we propose a novel tractography method that leverages Transformers to model the sequential nature of white matter streamlines, enabling the prediction of fiber directions by integrating both the trajectory context and current diffusion MRI measurements. To incorporate spatial information, we utilize CNNs that extract microstructural features from local neighborhoods around each voxel. By combining these complementary sources of information, our approach improves the precision and completeness of neural pathway mapping compared to traditional tractography models. We evaluate our method with the Tractometer toolkit, achieving competitive performance against state-of-the-art approaches, and present qualitative results on the TractoInferno dataset, demonstrating strong generalization to real-world data. Our code is publicly available at https://github.com/ItzikWaizman/TractoTransformer.

## 1   Introduction

Tractography is a key technique for analyzing diffusion-weighted imaging (DWI) data, aiming to reconstruct the complex 3D trajectories of white matter fibers—a fundamental step in understanding brain connectivity, development and neurological disorders [1, 2]. It exploits the principle that water molecules preferentially diffuse along axonal fibers, enabling the indirect estimation of fiber orientations from diffusion-weighted measurements acquired via diffusion magnetic resonance imaging (dMRI). Conceptually, tractography can be framed as a pathfinding problem: inferring plausible neural fiber pathways from noisy and ambiguous data while addressing challenges such as crossing, merging, and fanning fiber bundles. Traditional tractography methods rely on mathematical models that fit an estimated fiber orientation distribution function (fODF) to the measured DWI at each voxel, such as diffusion tensor imaging (DTI) [3], multi-tensor models [4], ball-and-sticks [5], Q-ball imaging (QBI) [6], and spherical deconvolution [7]. These orientation functions serve as local directional priors that guide the reconstruction of white matter pathways using deterministic, probabilistic, or combinatorial tracking strategies.

While classical tractography methods have significantly advanced our understanding of white matter architecture, they remain constrained by model-based assumptions—such as simplified representations of diffusion and voxel-wise independence [8]. These limitations have spurred the development of data-driven alternatives that learn directly from dMRI data [9]. Machine learning approaches offer greater flexibility in capturing complex white matter configurations, including fiber crossings and branchings, without imposing explicit assumptions about tissue properties or the dMRI signal.

39th Conference on Neural Information Processing Systems (NeurIPS 2025).

Although recent learning-based strategies have shown encouraging results, many still fall short of fully exploiting the underlying structure of the diffusion measurements, as they predict each voxel's orientation in isolation—disregarding either spatial dependencies [10, 11, 12, 13, 14] or the sequential structure of white matter tracts [15, 16, 17, 18, 19, 20]. Consequently, fiber orientation predictions tend to degrade in anatomically intricate or ambiguous regions.

In this work, we effectively leverage both the spatial and contextual information inherent in the data by proposing a spatio-sequential formulation of the fODF estimation task. Specifically, local features are first extracted from the dMRI volume using a 3D CNN, then passed to a decoder-only Transformer that predicts the fODF at each point along a streamline, conditioned on the preceding trajectory—offering a principled integration of fiber orientation features. Our contributions include:

- An algorithmic formulation of tractography as a pathfinding task, inspired by attention-based auto-regressive language models and spatially aware encoding.

- A tractography model that achieves state-of-the-art performance on a widely used benchmark, outperforming existing methods in key metrics.

- Open-source code infrastructure for training tractography models on multi-subject datasets, with support for in vivo diffusion MRI scans.

## 2 Related Work

In recent years, machine learning has emerged as a powerful tool for advancing tractography, moving beyond the limitations of traditional model-based approaches [21]. Early work by Neher et al. (2015, 2017) [10, 11] introduced a pioneering machine learning-based tractography method that uses a random forest (RF) classifier to guide streamline progression based on raw diffusion MRI data. This method demonstrated improved performance, particularly in complex fiber configurations, by taking advantage of data-driven decision-making to predict fiber directions and terminations.

Building on the idea of sequential data processing, Poulin et al. (2017) proposed LearnToTrack [12] and Benou et al. (2019) proposed DeepTract [13]. Both frameworks utilize recurrent neural networks (RNNs) for tractography, but differ in the way they frame the task. The former addressed streamline tractography as a regression problem by predicting continuous (deterministic) tracking directions, while the latter takes a classification approach by outputting a distribution over discrete directions on the unit sphere, thus allowing probabilistic tractography as well as deterministic. By treating streamlines as sequences of DWI data, RNN models capture the sequential dependencies of the data as context for inferring local fiber orientations. While RNNs enable sequential data processing, they are now often outperformed by Transformers, which offer better parallelization and long-range dependency handling.

Wegmayr et al. (2021) introduced Entrack [14], a probabilistic spherical regression approach that incorporates entropy regularization to manage uncertainty in fiber orientation estimation. Entrack uses the Fisher-von-Mises distribution to model the posterior distribution of local streamline directions, enhancing the robustness of the tractography in noisy conditions. This probabilistic approach is particularly well-suited for complex fiber architectures where multiple crossing fibers are present.

The exploration of reinforcement learning for tractography was advanced by Théberge et al. (2021) with the introduction of TrackToLearn [22]. This framework frames tractography as a reinforcement learning problem, where an agent learns to navigate white matter pathways by optimizing a reward function based on alignment with principal diffusion directions. This method does not require ground-truth tractograms for training, making it versatile across different datasets.

Hosseini et al. (2022) proposed CTtrack [23], a method combining CNNs and Transformers for fODF estimation. In CTtrack, a CNN projects diffusion MRI data to a lower-dimensional space, which is then processed by a Transformer to estimate fODFs as spherical harmonic coefficients. While both CTtrack and our proposed TractoTransformer combine CNNs and Transformers, their modeling paradigms differ fundamentally. CTtrack processes DWI data in a non-sequential manner, while our proposed TractoTransformer treats tractography as an auto-regressive sequence modeling task, offering a more structured and context-aware framework tailored to the intrinsic sequential nature of tractography.

# 3 Methodology

The main goal of the proposed TractoTransformer method is to extract streamlines—sequences of $(x, y, z)$ coordinates representing fiber pathways—from volumetric DWI data. The core concepts are illustrated in Figure 1. A detailed formulation and description of the data are provided in Section 3.1. The network architecture and its key components are presented in Section 3.2, while Section 3.3 outlines the training process for conditional fODFs prediction and the optimization strategies used to enhance model performance. Finally, Section 3.4 describes the inference procedure for streamline tracking on unseen data using the trained model.

## 3.1 Model Formulation

Streamline tractography aims to reconstruct white matter pathways by inferring plausible fiber trajectories from diffusion MRI data. We frame this problem as a sequential prediction task, where the likelihood of fiber orientations at a given point along a streamline is predicted based on the history of all previous DWI measurements along that path. Our dataset consists of DWI scans and the corresponding tractography data of $N$ subjects. For each subject, we have:

1. A 4D DWI volume $\boldsymbol{X} \in \mathbb{R}^{H \times W \times D \times G}$, where $H$,$W$, and $D$ are spacial dimensions, and $G$ corresponds to the number of magnetic field gradient directions applied during the dMRI scan. Axial views extracted from a volumetric DWI dataset of a single subject, acquired at six (out of 65) different gradient directions are shown in Figure 3.

2. A set of reference streamlines $\mathcal{S} = \left\{ \boldsymbol{s}^{(1)}, \ldots, \boldsymbol{s}^{(M)} \right\}$, representing a whole-brain tractography corresponding to $\boldsymbol{X}$, where each streamline $\boldsymbol{s}^{(m)} = (s_1^m, s_2^m, \ldots, s_{N_m}^m)$ is a sequence of 3D points in RAS (Right-Anterior-Superior) coordinates, commonly used in to standardize anatomical positions.

We feed our model with sequences of DWI values sampled along the coordinate path of a streamline. That is, given a streamline $\boldsymbol{s} = (s_1, \ldots, s_n)$, the input to the model is the sequence $\{\boldsymbol{X}(\boldsymbol{s}_1), \ldots, \boldsymbol{X}(\boldsymbol{s}_n)\}$.

At each point along a streamline, the model is trained to predict a conditional fODF, represented as a discrete probability distribution over a fixed set of $K + 1$ classes. Here, $K$ denotes a set of directions uniformly distributed on the unit sphere. Specifically, given a prefix trajectory $(s_1, \ldots, s_i)$, the output at point $s_i$ is:

$$\mathbb{P}(\boldsymbol{f} \mid \boldsymbol{X}(s_1), \ldots, \boldsymbol{X}(s_i)), \tag{1}$$

where $\boldsymbol{f} = (f_1, \ldots, f_K)$ is a discrete probability distribution over the direction classes defined by the spherical tessellation of $K = 724$ directions, along with an additional class representing end of fiber (EoF). This conditional formulation reflects the core assumption of our model: the fiber orientation at a given point depends not only on the local microstructural context (captured by the DWI signal), but also on the trajectory taken to reach that point.

## 3.2 Model Architecture

The proposed TractoTransformer leverages the strengths of both Transformers and convolutional neural networks (CNNs) to predict conditional fODFs from DWI data. Its architecture is illustrated in Figure 1.

**3D Input Embedding.** To embed the input sequence for the Transformer, we first enhance each voxel representation along a streamline by incorporating local spatial context using a 3D convolutional neural network (3D-CNN). For each point in the sequence, the 3D-CNN processes a surrounding voxel cube to extract microstructural features from the local diffusion signal. This step improves the voxel-wise representation and expands the effective receptive field, providing the model with spatial context. To reduce computational cost, the 3D-CNN is applied only to the batch of voxels corresponding to the current set of streamlines, rather than the entire brain volume.

The resulting spatially enhanced feature vectors serve as input tokens to the Transformer. To preserve the sequential order of streamline points, we apply standard sinusoidal positional encodings, as introduced by Vaswani et al. [24]. This enables the model to account for trajectory history, ensuring

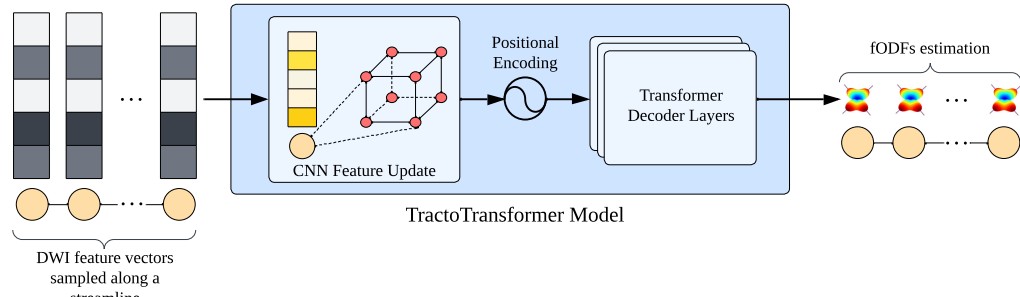

Figure 1: Overview of the TractoTransformer model framework. Streamlines are represented as sequences of DWI feature vectors, where each vector is derived from the raw dMRI data of a specific voxel, sampled using spherical harmonics. The TractoTransformer model consists of a 3D-CNN layer, positional encoder, and Transformer layers. The CNN is applied to each streamline voxel (represented by a diffusion measurement vector) and its nearby spatial neighbors. The entire encoded streamline is fed into the transformer. The model outputs predicted fODFs at each voxel, which are used to guide subsequent tractography.

that each orientation prediction is informed not only by local voxel features but also by the path taken to reach that point—an important consideration for anatomically plausible tractography.

**Decoder-Only Transformer.** We use a standard decoder-only Transformer architecture to process sequences of streamline data. Each decoder block includes masked multi-head self-attention and a position-wise feed-forward network, both followed by residual connections and layer normalization. A causal attention mask enforces autoregressive prediction by preventing access to future positions, while a padding mask blocks attention to invalid inputs. This design allows the model to capture long-range dependencies and contextual patterns along the streamline. The final output is mapped to the target space via fully connected layers, followed by a softmax function that yields a probability distribution over possible directions and an end-of-fiber (EoF) class. Further architectural and implementation details, including specific hyperparameters and training configurations, are provided in Section 4.

### 3.3 Model Optimization and Loss Function

We use the reference streamlines provided in the dataset to construct labels for supervised learning, training the model to predict conditional fODFs during sequence processing. For each reference streamline, direction vectors are computed between consecutive points and normalized to unit vectors. Since the output classes correspond to directions on the unit sphere and possess a geometric structure with well-defined angular distances, it is appropriate to weigh classification errors accordingly [13].

To this end, we construct a soft label distribution by smoothing each unit direction over the sphere using a Gaussian kernel. Formally, given a unit direction $\theta$ (i.e., the direction between two consecutive streamline points) and a set of unit directions $\{\alpha_i\}_{i=1}^{K}$ defining the discrete class space, we compute the angular distance $d_i$ between $\theta$ and each $\alpha_i$, and assign weights as:

$$w_i = \exp\left(-\frac{d_i^2}{2\sigma^2}\right),\tag{2}$$

where $\sigma$ is the standard deviation of the Gaussian kernel. The resulting soft label is a normalized probability distribution over directions:

$$y_{\text{smooth}}[i] = \frac{w_i}{\sum_{j=1}^{K} w_j}.\tag{3}$$

This distribution decays smoothly with increasing angular distance on the unit sphere, as illustrated in Figure 2, and is used to supervise the model's predictions.

Figure 2: Visualization of the smoothed label distribution on the unit sphere. The generated distribution decays as the distance on the unit sphere increases, providing a probabilistic framework for supervising the model's fODF predictions.

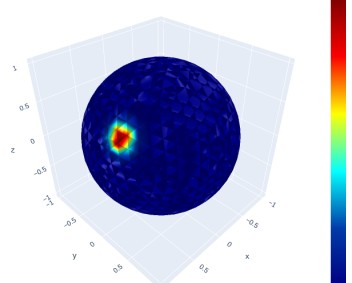

To train our TractoTransformer model, we employ the Kullback-Leibler divergence (KL-Div) loss to measure the divergence between the predicted discrete distribution and the corresponding smooth target label. Given a point $s_i$, a target label distribution $y_{\text{smooth}}$ associated with $s_i$, and a model prediction $y_{\text{pred}} = \mathbb{P}(\boldsymbol{f} \mid \boldsymbol{X}(s_1), \ldots, \boldsymbol{X}(s_i))$, the KL-Div is formally defined as:

$$\mathcal{L}_{\text{KL}}(y_{\text{smooth}}, y_{\text{pred}}) = \sum_{j=1}^{K} y_{\text{smooth}}[j] \log \left( \frac{y_{smooth}[j]}{\mathbb{P}\left(f_j \mid \boldsymbol{X}(s_1), \ldots, \boldsymbol{X}(s_i)\right)} \right), \tag{4}$$

The mean loss is computed at each prediction step along the streamline. The KL-Div loss quantifies the information loss incurred when $y_{\text{pred}}$ is used to approximate $y_{\text{true}}$, making it well-suited for evaluating the accuracy of probabilistic predictions against the ground truth distribution of fiber orientations. This loss function is particularly appropriate in scenarios where both the predicted outputs and the target labels are probability distributions, as it encourages the model to produce outputs that closely align with the empirical data.

### 3.4 Streamline Tractography Inference

Once the model is trained, tractography is initiated by sampling random seed points from the provided white matter mask, each defining the starting location of a fiber trajectory. Tractography proceeds iteratively: at each step, the model receives the current point along with the accumulated tracking history and predicts a conditional fODF, auto-regressively conditioned on previously generated points in the streamline. This design ensures that each orientation prediction incorporates both local features and the full trajectory context, capturing the sequential dependencies inherent in white matter pathways.

The tracking direction is selected as the one with the highest probability in the predicted fODF, resulting in a deterministic propagation scheme. However, unlike classical deterministic methods, our predictions are context-aware—conditioned on the entire streamline history—enabling robust direction selection even in anatomically challenging regions such as fiber crossings or areas of high uncertainty.

After selecting a direction, the streamline is advanced by a fixed step in RAS space, the new point is appended to the trajectory, and the process is repeated until a stopping criterion is met:

1. The class chosen from the prediction of the model is EoF class.

2. The next step is outside of the bounds of the MRI image.

3. The next step is outside of the white matter mask.

4. The angle between two consecutive steps exceeds a predefined threshold.

5. The fractional anisotropy (FA) values in the next step are less than a predefined threshold.

The collection of generated trajectories constitutes a set of approximate streamlines which together form the final tractogram. This process is detailed in the pseudocode provided in Algorithm 1.

**Algorithm 1** Streamline Tractography Algorithm

---

**Require:** Trained model, white matter mask, seed points, stopping criteria
**Ensure:** Tractogram of streamlines
1: **for** each seed point **do**
2:     Initialize streamline with seed point
3:     **while** stopping criteria are not met **do**
4:         Feed the current streamline into the model
5:         Get conditional fODFs from the model
6:         Select direction as `argmax` of conditional fODF
7:         Compute next point by stepping in the selected direction
8:         **if** next point satisfies stopping criteria **then**
9:             Terminate streamline
10:        **else**
11:            Add next point to the streamline
12:        **end if**
13:     **end while**
14:     Store the completed streamline in tractogram
15: **end for**

---

## 4 Experiments

### 4.1 Datasets

For this study, we used two publicly available tractography datasets. The first is the ISMRM 2015 Tractography Challenge phantom dataset [25], which has been one of the most widely used benchmarks in the field over the past decade. It contains a high-quality 4D DWI volume with dimensions $90 \times 108 \times 90 \times 100$ after resampling, along with a comprehensive set of 270,000 ground truth white matter streamlines.

The second dataset is TractoInferno [26], the largest open-source, multi-site tractography dataset to date, comprising diffusion data from 286 subjects. The dataset is partitioned into 198 subjects for training, 60 for validation, and 28 for testing. Each subject includes a 4D diffusion-weighted imaging (DWI) volume with higher spatial resolution than the ISMRM dataset. Although the exact dimensions vary between subjects (for example, the test subject whose tractography is shown in Figure 5 has a spatial volume of $141 \times 184 \times 120$), the number of gradient directions also varies, ranging from 22 to 132, and is resampled to 100 directions for consistency. In addition, the dataset provides rich ground-truth tractography, averaging over 1 million streamlines per subject. These streamlines are generally shorter than those in the ISMRM dataset and, after resampling to a constant step size of $3\,\mathrm{mm}$, can contain up to 100 3D points.

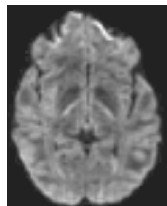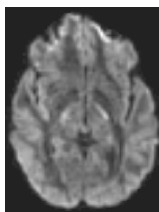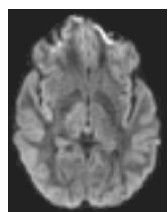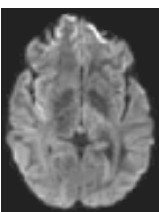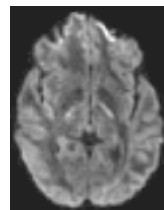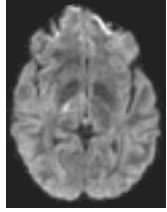

Figure 3: Axial slices from a volumetric DWI dataset of a single subject where each scan was acquired at a different diffusion gradient direction. Data source: sub-1024 DWI from TractoInferno dataset [26]

.

### 4.2 Preprocessing.

To ensure consistency across subjects and reduce variability arising from acquisition protocols, we apply several preprocessing steps. First, we represent the DWI signal using spherical harmonic

coefficients sampled over a fixed set of gradient directions. This step addresses inter-subject variability in gradient schemes and provides a standardized input format for the model. Next, we resample the reference streamlines to maintain a consistent step size between consecutive points in the right-anterior-superior (RAS) space, ensuring uniform spatial resolution across all samples and supporting reliable modeling and downstream analysis. Finally, we augment the dataset by reversing streamline orientations, increasing data diversity, and enabling the model to learn more robust features.

### 4.3 Implementation Details

**Architecture.** The model configurations are as follows. For the 3D-CNN component, we use a single 3D convolutional layer with a kernel size of $3 \times 3 \times 3$. We evaluated two input variants: (i) the original diffusion-weighted imaging (DWI) signals, resampled to 100 fixed gradient directions, and (ii) spherical harmonic (SH) coefficients of order 12, representing the diffusion signal in the frequency domain. To mitigate overfitting, dropout with a probability of $p = 0.1$ was applied to all layers. The Transformer-based network consists of 8 decoder layers, each with 10 attention heads. Every decoder block includes a feed-forward network (FFN) with a hidden dimension of 512. The final Transformer output is passed through an additional FFN, projecting it to a 725-dimensional vector representing 724 candidate directions on the unit sphere and one end-of-fiber (EoF) class used to signal streamline termination.

**Training.** All models were trained using the Adam optimizer [27] with an initial learning rate of 0.005. Learning rate decay was applied by multiplying the rate by 0.7 if the accuracy did not improve by at least 0.3 over two consecutive epochs. For label smoothing, we employed a Gaussian kernel with a standard deviation of $\sigma = 0.1$. Target labels were represented as discrete probability distributions over 725 classes ($K = 724$), corresponding to an angular resolution of approximately $3.5°$. Training was conducted for 30 epochs with a batch size of 20, using up to four NVIDIA V100 GPUs with 32GB of memory each.

**Inference.** We used an angular threshold of 70 degrees and an FA threshold of 0.05. The step size was set to 1 mm for the ISMRM dataset and 3 mm for TractoInferno, matching each dataset's native spatial resolution. Streamline generation was performed using approximately 200,000 seed points for ISMRM and about one million for TractoInferno, processed in batches of 100 and distributed evenly across four GPUs to leverage data-parallel tractography. To further optimize inference, we employ key–value (KV) caching in the Transformer decoder to reuse attention states from previous steps, substantially reducing inference time and memory overhead. On average, full-brain tractography for a single subject required approximately 43 minutes on four NVIDIA V100 GPUs.

### 4.4 Evaluation on the Synthetic ISMRM Dataset

#### 4.4.1 Whole-Brain Tractography Evaluation

To evaluate our model, we trained it on the ISMRM dataset using an 80/20 split of reference streamlines for training and validation. Training took 12 hours. Whole-brain tractography was then performed by seeding from random points within the white matter mask. For comprehensive benchmarking and to facilitate future comparisons, we report our results using both the classic Tractometer [28] (2015 edition) and the updated Tractometer [29] (2023 edition), the latter incorporating ROI-based segmentation to improve reliability and reproducibility.

We report four key metrics: **valid connection (VC)** (valid connection rate), **overlap (OL)** (overlap with ground truth), **overreach (OR)** (overreach beyond anatomical boundaries), and the **F1 score**, which balances precision and recall. Results in Table 1 demonstrate that TractoTransformer outperforms state-of-the-art tractography methods. The spherical harmonics input configuration achieves the highest overall performance, with a VC of 84%, an overlap of 79%, and the best F1 score (75%). These results indicate accurate and specific reconstruction of white matter connections, with high sensitivity and relatively low overreach (27%).

#### 4.4.2 Complex Fiber Bundles Reconstruction

To further demonstrate the advantages of our history-aware streamline propagation, we compare the bundle-specific reconstruction of TractoTransformer (with DWI-input configuration) with the deterministic tractography algorithm implemented in the MITK Diffusion toolbox. The MITK method

Table 1: Comparison of the proposed TractoTransformer performance with that of state-of-the-art methods on the TractoInferno dataset. For each metric, the best results are shown in **bold** and the second-best are underlined. Our method achieves the top performance in VC, OL, and F1. Entries marked with an asterisk (*) correspond to results obtained using the 2023 Tractometer edition.

| Model | VC (%)↑ | OL (%)↑ | OR (%)↓ | F1 (%)↑ |
|---|---|---|---|---|
| **ISMRM Mean** | 54 | 31 | 23 | 44 |
| **RF [10, 11]** | 67 | 75 | 31 | - |
| **LearnToTrack [12]** | 42 | 64 | 35 | 64 |
| **DeepTract [13]** | 71 | 69 | 23 | 70 |
| **Entrack [14]** | 65 | 60 | 36 | 58 |
| **Track-to-learn [22]** | 68 | 62 | - | - |
| **CTtrack [23]** | 57 | 50 | **16** | 60 |
| **TractoTransformer** | 82 | **82** | 35 | 71 |
| **TractoTransformer SH input** | **84** | 79 | 27 | **75** |
| **TractoTransformer*** | 82 | 84 | 31 | 75 |
| **TractoTransformer SH input*** | 82 | 81 | 26 | 78 |

reconstructs streamlines by following local diffusion maxima, without accounting for trajectory history or incorporating global contextual information. We focus on complex white matter bundles characterized by extensive fiber crossings and branchings, such as the Right Brainstem Pontine tract and the Left Cingulum bundle. As shown in Table 2, TractoTransformer achieves markedly higher valid connection counts, overlap, and F1 scores in these challenging bundles. Figure 4 illustrates qualitative differences between the two methods, highlighting the enhanced anatomical plausibility achieved by TractoTransformer reconstructions.

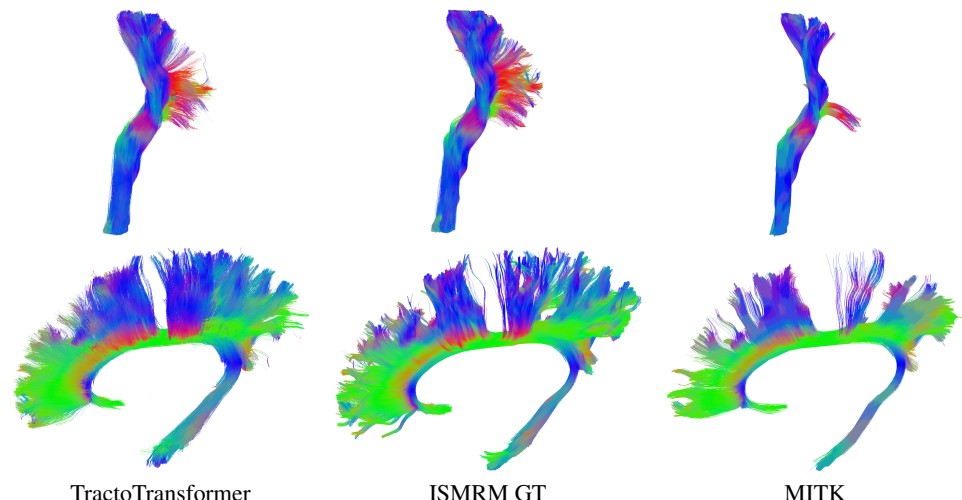

| TractoTransformer | ISMRM GT | MITK |

Figure 4: Visual comparison of tractography results in regions with complex fiber architecture from the ISMRM dataset. **Top:** Right Brainstem Pontine tract. **Bottom:** Left Cingulum bundle. Each column shows reconstructions obtained with the proposed TractoTransformer, the ISMRM ground truth, and the MITK deterministic approach.

## 4.5 Ablation Study

Table 3 summarizes ablation results, highlighting the contribution of each component in our framework. Whole-brain tractography was evaluated on the ISMRM dataset after removing the 3D-CNN module, reverse streamline augmentation, or label smoothing. All experiments used the TractoTransformer variant with raw DWI input and were quantitatively assessed using the 2023 Tractometer toolkit [29].

Table 2: Quantitative comparison between MITK and TractoTransformer on complex bundles. VC denotes the number of valid connections.

| Bundle | VC↑ | OL (%)↑ | OR (%)↓ | F1 (%)↑ |
|---|---|---|---|---|
| **Right Brainstem Pontine Tract** | | | | |
| MITK | 11103 | 39.48 | **18.37** | 53.22 |
| TractoTransformer | **31996** | **88.31** | 27.36 | **79.71** |
| **Left Cingulum Bundle** | | | | |
| MITK | 7619 | 46.97 | **31.71** | 55.66 |
| TractoTransformer | **16792** | **89.62** | 39.09 | **72.52** |

Table 3: Ablation study of the TractoTransformer framework evaluated for the ISMRM data with the Tractometer toolkit (2023 edition).

| Model | VC (%)↑ | OL (%)↑ | OR (%)↓ | F1 (%)↑ |
|---|---|---|---|---|
| **TractoTransformer** | **81.51** | **83.72** | **30.83** | 74.78 |
| **-3D-CNN** | 69.86(-11.65) | 80.70(-3.02) | 32.84(+2.01) | 72.66(-2.12) |
| **-Reverse Streamlines** | 79.81(-1.70) | 82.94(-0.78) | 33.08(+2.25) | 73.76(-1.02) |
| **-Smooth Labels** | 79.77(-1.74) | 82.81(-0.91) | 30.86(+0.03) | 74.85(+0.07) |

The largest performance drop is observed when excluding the 3D-CNN module, which reduces VC by 11.65% and slightly lowers the F1 score, underscoring the importance of local spatial context for accurate trajectory estimation. Removing reverse streamline augmentation leads to a smaller decline, indicating that directional diversity supports regularization but is less critical. Omitting label smoothing has minimal effect on F1, with only slight decreases in VC and OL. Overall, while all components contribute, the 3D-CNN module remains essential for anatomically plausible reconstructions.

### 4.6 Evaluation on the In-vivo TractoInferno Dataset

To evaluate our method on in-vivo dMRI data, we used the TractoInferno dataset [26]. Due to computational constraints, training was performed on ten subjects, validation on two, and testing on four. The architecture and training setup matched those used for the ISMRM dataset, except for multi-subject training. Training took approximately seven days. Figure 5 presents whole-brain and bundle-level reconstructions for one test subject (sub-1019), showing TractoTransformer's ability to generalize across subjects and recover complex fiber pathways. Table 4 reports the average quantitative results across the four test subjects. All baselines were implemented and evaluated by the TractoInferno authors using their official pipeline. TractoTransformer achieved the highest Overlap and Dice (F1) scores, demonstrating a superior balance between anatomical coverage and precision. The full per-subject results are provided in the Appendix, along with a Pareto visualization illustrating the overlap-overreach tradeoff of the compared methods. The Pareto analysis indicates that TractoTransformer lies above the baseline Pareto front, achieving an average overlap gain of 0.027 (∼4.8%) at comparable overreach levels.

Table 4: Average performance across four TractoInferno subjects. Our method achieves the highest Dice and Overlap, indicating a superior balance between coverage and precision.

| Metric | Det-Cosine | Det-SE (Learn2Track) | Prob-Sphere (DeepTract) | Prob-Gaussian (SOTA) | Prob-Mixture | TractoTransformer (Ours) |
|---|---|---|---|---|---|---|
| Dice↑ | 0.609 | 0.575 | 0.596 | 0.612 | 0.391 | **0.628** |
| Overlap↑ | 0.549 | 0.495 | 0.540 | 0.578 | 0.292 | **0.589** |
| Overreach↓ | 0.243 | 0.191 | 0.219 | 0.288 | **0.058** | 0.263 |

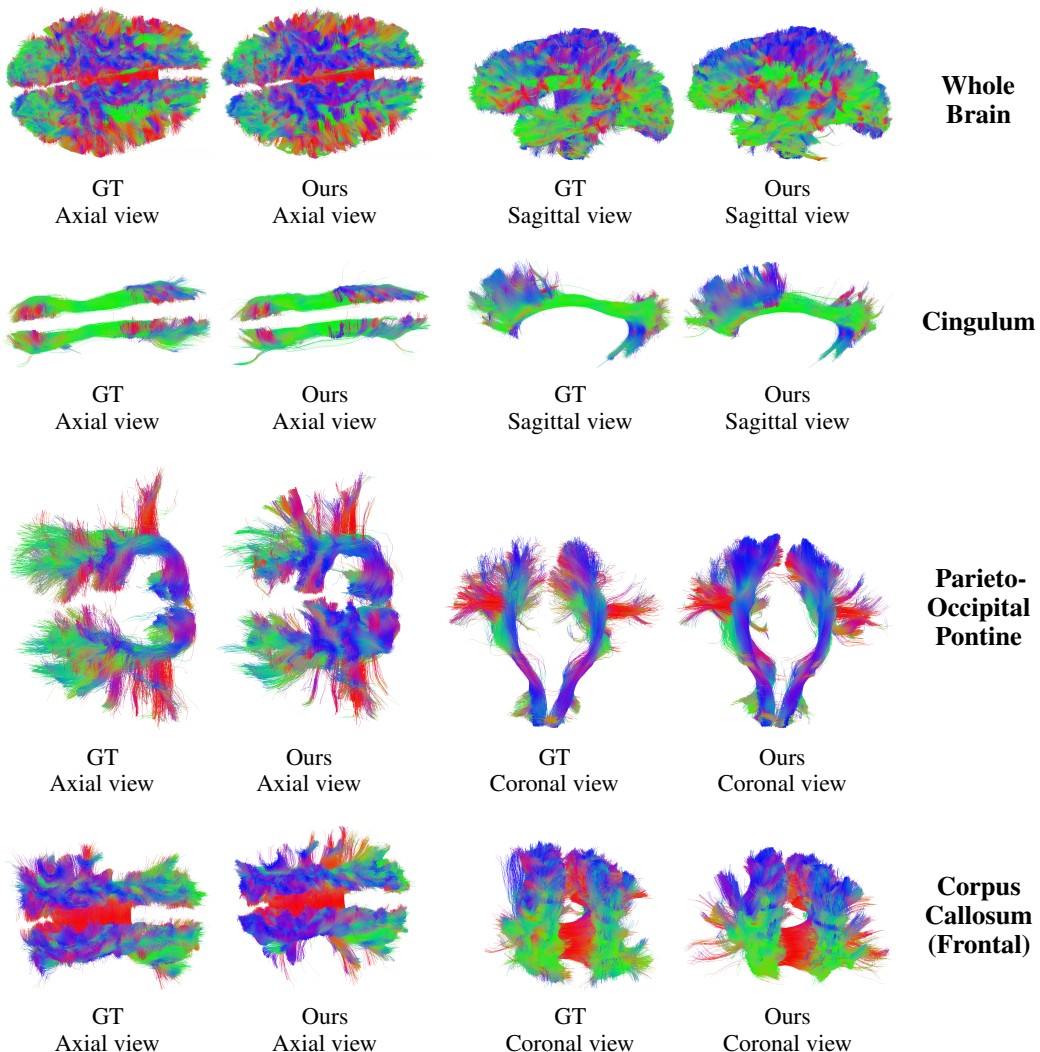

**Whole Brain**

**Cingulum**

**Parieto-Occipital Pontine**

**Corpus Callosum (Frontal)**

Figure 5: Visual comparison of tractography outputs from TractoInferno (GT) and TractoTransformer (TT) across four regions. Each row shows a different tract, with matched views from both models.

## 5   Conclusions, Limitations, and Broader Impact

**Conclusions.** We presented *TractoTransformer*, a hybrid CNN–Transformer framework for diffusion MRI tractography that integrates local microstructural context with sequential trajectory modeling. By leveraging the prefix trajectory to guide the tracking process, the model produces anatomically accurate reconstructions and effectively resolves complex fiber configurations such as crossing and kissing bundles. Comprehensive evaluations on the ISMRM and TractoInferno datasets, including whole-brain and per-bundle analyses against classical and deep learning baselines, demonstrate the superior performance of the proposed TractoTransformer across the tested benchmarks.

**Limitations.** The main bottlenecks are computational, arising from high memory usage and relatively long inference time, which may limit scalability in large or high-resolution datasets. Efficiency could be improved through optimized attention mechanisms (e.g., FlashAttention) or model compression to reduce resource demands and enable broader applicability.

**Broader Impact.** *TractoTransformer* provides a high-performing and accessible framework for AI-driven tractography, with potential applications in both neuroscience research and clinical practice. Its modular design, together with publicly available code and pretrained models, promotes reproducibility and ease of adoption, enabling researchers to extend and advance data-driven neuroimaging.

## Acknowledgments and Disclosure of Funding

We thank the authors of the TractoInferno dataset [26] for their courtesy in sharing per-subject evaluation scores of the implemented baselines, which enabled the quantitative analysis presented in the appendix. This work was partially supported by the Israel Ministry of Science and Technology (T.R.R MOST 0007311/1001817977).

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

# A    Additional Results

Table 5: Performance comparison across four subjects using Dice, Overlap, and Overreach metrics for each method. All baseline methods were implemented and evaluated by the TractoInferno authors. Our method achieves the highest Dice score on three out of four subjects and the second-best on the remaining one.

|  |  | Det-Cosine | Det-SE (Learn2Track) | Prob-Sphere (DeepTract) | Prob-Gaussian (SOTA) | Prob-Mixture | TractoTransformer (Ours) |
|---|---|---|---|---|---|---|---|
| **sub-1006** | dice↑ | 0.618 | 0.544 | 0.570 | 0.597 | 0.399 | **0.626** |
|  | overlap↑ | 0.575 | 0.470 | 0.525 | **0.593** | 0.284 | 0.590 |
|  | overreach↓ | 0.281 | 0.231 | 0.248 | 0.381 | **0.074** | 0.262 |
| **sub-1019** | dice↑ | 0.614 | 0.558 | 0.597 | 0.606 | 0.381 | **0.654** |
|  | overlap↑ | 0.551 | 0.493 | 0.540 | 0.566 | 0.266 | **0.625** |
|  | overreach↓ | 0.239 | 0.198 | 0.227 | 0.266 | **0.051** | 0.279 |
| **sub-1024** | dice↑ | 0.585 | 0.593 | 0.600 | **0.624** | 0.354 | 0.602 |
|  | overlap↑ | 0.511 | 0.493 | 0.533 | **0.579** | 0.302 | 0.562 |
|  | overreach↓ | 0.196 | 0.153 | 0.192 | 0.245 | **0.039** | 0.267 |
| **sub-1061** | dice↑ | 0.618 | 0.606 | 0.616 | 0.620 | 0.430 | **0.629** |
|  | overlap↑ | 0.560 | 0.523 | 0.562 | 0.575 | 0.314 | **0.580** |
|  | overreach↓ | 0.256 | 0.180 | 0.209 | 0.259 | **0.068** | 0.244 |

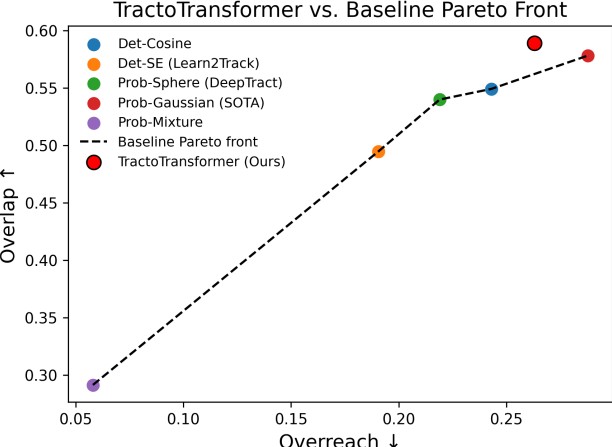

Figure 6: Pareto analysis of the overlap–overreach trade-off between TractoTransformer and baseline methods. The dashed line represents the Pareto front computed from the baselines, while the red marker denotes *TractoTransformer* (ours), which lies $0.027$ units above the baseline front, equivalent to a $4.8\%$ higher overlap at comparable overreach.

