# OpenReview forum: "TractoTransformer: Diffusion MRI Streamline Tractography using CNN and Transformer Networks"
_NeurIPS.cc/2025/Conference — NeurIPS 2025 poster_

### Official Review · Reviewer_LQ4s · 2025-06-20

**Clarity:** 3
**Significance:** 3
**Originality:** 3
**Rating:** 5
**Confidence:** 4

**Summary:**

The manuscript proposed a tractography algorithm based on machine learning, more specifically CNN and Transformer Architecture. The goal is to add more anatomical features (landmarks) and priors (path history) into the tractography process. Framing tractography as an sequence modeling task directly addresses a key limitation of many tractography methods (e.g predicting fiber orientations in a voxel without considering the path taken to get there).

**Questions:**

For table 1, are the reported results taken for their original publication? Are they reimplementation of the methods that you ran? Or are they standard run (from the original code + container) that you ran? This information would be important for context (default or tuned parameters, weights version, etc.)

The paper is transparent about the limitations: the 3D-CNN is memory-intensive, and inference takes 7 hours per subject (does resolution impact that? How many streamlines generated? What step-size?). This is a clear contrast with other ML tractography algorithm and possibly a barrier to practical use. While acknowledged the review should explore how this might be improved in future work.

**Ethical Concerns:**

["NO or VERY MINOR ethics concerns only"]

**Final Justification:**

I believe the author adressed the limitations I observed as well as the other reviewers feedback. Considering my initial review was 'optimistic', I think my initial rating should not change, but the changes to the manuscript and clarifications are in line with what I was expecting.

**Paper Formatting Concerns:**

Well written and well organized, but the figures' caption need more details and the authors should guide the reader focus (figure 3 or 4 look very similar between column, it is hard to know what to look for). I believe a lot more guidance is needed to highlight what we should see between column.

**Quality:**

3

**Strengths And Weaknesses:**

The combination of a 3D CNN for spatial context and a Transformer for sequential context is well-justified. The manuscript demonstrates good performance on the ISMRM 2015 tractography challenge benchmark  and shows interesting results on the more complex, real-world TractoInferno dataset. The ablation study demonstrates the contribution of each model module (CNN, reverse streamlines, smooth labels), supporting the authors design choices.

The qualitative evaluation on TractoInferno is very promising. However, the model was trained on only ten subjects from this dataset validated on two and tested on one. How well does this represent the full diversity of the 198 training subjects? This might limit the claimed generalization capability. I believe a simpler experiment on data such as the HCP test-retest (44x2 subject) for future work could help with a simpler hypothesis, it is expected that a good algorithm should have have high overlap of bundles (requires a segmentation methods). I recommend this script for comparing bundles (in the same space, co-registered): https://github.com/scilus/scilpy/blob/master/scripts/scil_bundle_pairwise_comparison.py
And this script to compare whole brain tractogram (co-registered): https://github.com/scilus/scilpy/blob/master/scripts/scil_tractogram_pairwise_comparison.py

---

> ### Author Rebuttal · Authors · 2025-07-29
>
> **Response to Reviewer LQ4s**
>
> Thank you for your thoughtful and constructive feedback, especially your recognition of our conceptual framing, ablation studies, and qualitative results. Your practical suggestions and questions are highly appreciated and have helped us better communicate the scope and limitations of our work.
>
> **Addressing Main Weaknesses**
>
> **1. Limited training/testing subset from TractInferno; generalization capability:**
> We acknowledge the limitation of training and testing on a relatively small subset of the TractInferno dataset, which was due to hardware constraints. Specifically, loading more than ten volumes was not feasible, and expanding further, even with methodological adjustments, would have significantly increased computational demands. While tractography results typically show limited qualitative variation across subjects, we agree that broader evaluation is important for demonstrating generalizability.
>
> To address this, we have extended our evaluation to include quantitative tractography results on four subjects from the TractInferno test set. For fair and standardized comparison, we contacted the TractInferno authors, who kindly provided us with official evaluation metrics: Dice, Overlap, and Overreach, for five leading baseline methods: Det-Cosine, Det-SE (Learn2Track), Prob-Sphere (DeepTract), Prob-Gaussian, and Prob-Mixture. These per-subject results were not published in the original TractInferno paper (which only reported averaged metrics), so these scores were shared with us upon request. For full implementation details of these baselines, we refer to the original TractInferno paper.
>
> As shown below, our method achieves the highest Dice score on three out of four subjects (sub-1006, sub-1019, and sub-1061), and the second-highest Dice on sub-1024. Importantly, these results were obtained using a model trained on only 10 subjects, compared to 198 used for the TractInferno baselines—highlighting the strong generalization capability of our approach, even with limited training data.
>
> Scaling the evaluation to the full TractInferno test set was not feasible at submission due to hardware constraints. However, to support reproducibility and enable future benchmarking, we are releasing our full codebase as open source. We believe these added quantitative results already provide meaningful evidence of our method’s effectiveness, and we encourage future work, especially from groups with greater computational resources, to build upon our approach and validate it at larger scale.
>
> **Table: Performance comparison (Dice, Overlap, Overreach) for each method on four TractInferno subjects**
>
> | **Subject** | **Metric**   | **Det-Cosine** | **Det-SE (Learn2Track)** | **Prob-Sphere (DeepTract)** | **Prob-Gaussian** | **Prob-Mixture** | **TractoTransformer** |
> |-------------|--------------|----------------|--------------------------|-----------------------------|-------------------|------------------|-----------------------|
> | sub-1006    | Dice         | 0.618          | 0.544                    | 0.570                       | 0.597             | 0.399            | **0.626**             |
> |             | Overlap      | 0.575          | 0.470                    | 0.525                       | **0.593**         | 0.284            | 0.590                 |
> |             | Overreach    | 0.281          | 0.231                    | 0.248                       | 0.381             | **0.074**        | 0.262                 |
> | sub-1019    | Dice         | 0.614          | 0.558                    | 0.597                       | 0.606             | 0.381            | **0.654**             |
> |             | Overlap      | 0.551          | 0.493                    | 0.540                       | 0.566             | 0.266            | **0.625**             |
> |             | Overreach    | 0.239          | 0.198                    | 0.227                       | 0.266             | **0.051**        | 0.279                 |
> | sub-1024    | Dice         | 0.585          | 0.593                    | 0.600                       | **0.624**         | 0.354            | 0.602                 |
> |             | Overlap      | 0.511          | 0.493                    | 0.533                       | **0.579**         | 0.302            | 0.562                 |
> |             | Overreach    | 0.196          | 0.153                    | 0.192                       | 0.245             | **0.039**        | 0.267                 |
> | sub-1061    | Dice         | 0.618          | 0.606                    | 0.616                       | 0.620             | 0.430            | **0.629**             |
> |             | Overlap      | 0.560          | 0.523                    | 0.562                       | 0.575             | 0.314            | **0.580**             |
> |             | Overreach    | 0.256          | 0.180                    | 0.209                       | 0.259             | **0.068**        | 0.244                 |
>
> **2. Test-retest reliability and suggested comparison scripts:**
> Thank you for highlighting the importance of test-retest validation and for recommending specific resources. We agree that evaluation on HCP test-retest data is an excellent way to benchmark robustness and will reference this in our discussion as a priority for future work. By providing our code and weights, we facilitate such analyses by both our team and the broader community.
>
> **Response to Specific Questions**
>
> **1. Table 1 Benchmarking:**
> All values in Table 1 are quoted directly from the original publications. We appreciate your request for clarity and will explicitly note this in our revisions to avoid ambiguity for readers.
>
> **2. Practical limitations: runtime, memory, resolution, streamline count, and step size:**
> Inference time and resource consumption are influenced by several factors: (a) Step size: 1mm for ISMRM, 3mm for TractInferno, matching each dataset’s reference sampling. Finer step sizes increase computation; (b) Streamline count: We match reference tractograms ($\sim$ 200,000 for ISMRM, $\sim$ 1,000,000 for TractInferno); (c) Resolution: We use native full resolution to preserve anatomical detail, which increases computational cost but benefits qualitative accuracy. (d) 3D-CNN design: The 3×3×3 kernel size was chosen to balance anatomical context and memory consumption; larger/deeper models exceeded our GPU capability. We agree these constraints represent an important direction for practical improvement. Notably, since streamlines are generated independently, inference runtime can be reduced almost linearly simply by parallelizing across computing units. We will explicitly highlight these practical limitations and the opportunities for acceleration in the revised manuscript.
>
> We appreciate the reviewer’s constructive comments and suggestions, which have helped us improve both the clarity and scope of our work. Thank you again for your thoughtful review and support.

---

### Official Review · Reviewer_Nf5j · 2025-06-29

**Clarity:** 3
**Significance:** 2
**Originality:** 2
**Rating:** 3
**Confidence:** 4

**Summary:**

The authors propose a new spatio-sequential tractography algorithm that incorporates spatial information along a predicted streamline. During training, the method leverages a set of ground truth streamlines and, at each point along a streamline, predicts an orientation probability to reach the next point. At inference time, the method can generate novel streamlines in a sequential manner. The approach is validated on two datasets and demonstrates both quantitatively and qualitatively plausible tractograms.

**Questions:**

* **Q1**: How are the reference streamlines generated and filtered?
* **Q2**: Why was 3×3×3 chosen, and how does that choice affect performance?
* **Q3**: Can the CNN be applied once to the full volume to avoid recomputing features at every streamline step?
* **Q4**: How is interpolation performed when sampling DWI values at off-grid points along a streamline?
* **Q5**: Does the model predict a full fODF per voxel, or a direction-conditioned distribution tied to the current streamline (closer to an orientation probability)?
* **Q6**: Which stopping criteria are learned? Please provide statistics on end-of-fiber (EoF) predictions versus other stopping causes.
* **Q7**: Why not input spherical-harmonic coefficients directly, given that the convolution is blind to spherical geometry?
* **Q8**: Can you identify regions where classical trackers fail but your model succeeds, and vice versa?
* **Q9**: Can the quantitative metric from the Tractometer dataset be used on the in-vivo dataset as well?

**Ethical Concerns:**

["NO or VERY MINOR ethics concerns only"]

**Final Justification:**

The rebuttal is strong, providing a deeper analysis of the strengths and weaknesses of the method, as well as useful practical clarifications. I am raising my score accordingly. However, the current draft of the paper requires major revisions to incorporate these new analyses, which would necessitate another round of review.

**Limitations:**

Yes.

**Quality:**

3

**Strengths And Weaknesses:**

**Strengths**:

* **End-to-end learning of tracking decisions**: Predicting the next streamline direction directly from local DWI patches removes the need for intermediate fODF estimation and may capture contextual cues that classical pipelines miss.
* **Spatial context included**: Even with a simple 3×3×3 voxel cube, the model incorporates local neighbourhood information, improving direction prediction over single-voxel inputs.
* **Preliminary quantitative gains**: Reported Tractometer scores and qualitative in-vivo bundles suggest performance that is competitive with or better than existing baselines.

**Weaknesses**:

* **Training–test leakage**: The Tractometer experiment uses the same diffusion volume for both training and evaluation; overlap between streamlines likely inflates the reported metrics.
* **Limited anatomical stress-testing**: The method’s performance in regions with complex fibre configurations (e.g., crossings, kissings, or fans) is not directly compared to existing fODF-based approaches.
* **Sparse methodological detail**: Important implementation details—such as step size, cube size reasoning, SH decomposition order, and the data split strategy—are not clearly presented.
* **Tiny convolutional “network”**: A single 3D convolution layer amounts to little more than spatial smoothing; there is no ablation to test whether deeper CNNs or SH coefficients as input would improve results.
* **Dependence on teacher streamlines**: The model is trained on labels from a single tracking algorithm, meaning any bias or failure in that method is directly inherited.

---

> ### Author Rebuttal · Authors · 2025-07-30
>
> **Response to Reviewer Nf5j**
>
> We thank the reviewer for the thorough and thoughtful review. Below, we address your main concerns and specific questions.
>
> **1. Training–test leakage**
>
> We acknowledge that our evaluation on the ISMRM 2015 phantom uses the same diffusion scan for training and testing. While this is a known limitation of the benchmark, it remains standard practice to enable fair, like-for-like comparisons across methods.
>
> Importantly, our model is context-aware: its predictions are based on both the local DWI and the full streamline path up to the current point. Because test streamlines are seeded randomly, even if they traverse seen voxels, the input trajectory differs—preventing simple memorization.
>
> To directly address the generalization concern, we conducted additional inference on four unseen TractInferno test subjects (sub-1006, 1019, 1024, 1061). We compared our model to five baselines implemented by the TractInferno authors, who kindly shared the per-subject results. We used their official evaluation script to compute Dice, Overlap, and Overreach metrics.
> These comparisons were not included in the original publication, as TractInferno reports only dataset-wide averages, which we were unable to reproduce due to hardware constraints. Notably, our model was trained on only 10 subjects, whereas the TractInferno baselines were trained on 198. Despite this, as shown in Table 1, our method achieves the highest Dice score on three of the four subjects and the second-best on the remaining one, demonstrating strong generalization from limited training data.
>
> **Table 1: TractInferno test subjects — Dice, Overlap, Overreach**
> | Subject   | Metric    | Det-Cosine | Det-SE | Prob-Sphere | Prob-Gaussian | Prob-Mixture | TractoTransformer |
> |-----------|-----------|------------|--------|--------------|----------------|---------------|-------------------|
> | sub-1006  | Dice      |0.618|0.544|0.570|0.597|0.399| **0.626**|
> |           | Overlap   |0.575|0.470|0.525|**0.593**|0.284| 0.590|
> |           | Overreach |0.281|0.231|0.248|0.381|**0.074**| 0.262|
> | sub-1019  | Dice      |0.614|0.558|0.597|0.606|0.381| **0.654**|
> |           | Overlap   |0.551|0.493|0.540|0.566|0.266| **0.625**|
> |           | Overreach |0.239|0.198|0.227|0.266|**0.051**| 0.279|
> | sub-1024  | Dice      |0.585|0.593|0.600|**0.624**|0.354| 0.602|
> |           | Overlap   |0.511|0.493|0.533|**0.579**|0.302| 0.562|
> |           | Overreach |0.196|0.153|0.192|0.245|**0.039**| 0.267|
> | sub-1061  | Dice      |0.618|0.606|0.616|0.620|0.430| **0.629**|
> |           | Overlap   |0.560|0.523|0.562|0.575|0.314| **0.580**|
> |           | Overreach |0.256|0.180|0.209|0.259|**0.068**| 0.244|
>
>
> **2. Limited anatomical stress-testing**
>
> In order to address this concern, we conducted a comparative experiment against MITK Diffusion, a deterministic tractography pipeline that follows voxelwise principal directions derived from fODFs, without incorporating streamline history or global context.
>
> We used the ISMRM phantom and evaluated two challenging bundles—Right Brainstem Pontine and Left Cingulum—which exhibit crossings and branching. Table 2 compares *number* of valid connections (VC) using the official Tractometer tool. Table 3 compares *percentage* metrics for whole-brain tracking.
> Our model clearly outperforms in terms of valid connections, overlap, and F1 score, highlighting its effectiveness in detecting these complex fiber bundles. To further highlight these differences, we will include explicit visual comparisons of bundle reconstructions along with these quantitative results in the revised manuscript.
>
> **Table 2: MITK vs. TractoTransformer on complex bundles (ISMRM, Tractometer)**
>
> | **Bundle**                    | **Method**        | **VC** | **OL (%)** | **OR (%)** | **F1 (%)** |
> |-------------------------------|-------------------|--------|------------|------------|------------|
> | Right Brainstem Pontine Tract |MITK|11103|39.48|**18.37**| 53.22|
> |                               |TractoTransformer|**31996**|**88.31**|27.36|**79.71**|
> | Left Cingulum Bundle          |MITK|7619|46.97|**31.71**|55.66|
> |                               |TractoTransforme|**16792**|**89.62**|39.09|**72.52**|
>
> **Table 3: Whole-brain tractography (ISMRM, Tractometer)**
>
> | **Method**         | **VC (%)** | **OL (%)** | **OR (%)** | **F1 (%)** |
> |--------------------|------------|------------|------------|------------|
> | MITK               |55.74|48.03|**28.56**|54.98|
> | TractoTransformer  |**81.51**|**83.72**|30.83|**74.78**|
>
> **3. Sparse methodological details**
>
> Thank you for noting this. We will clarify all implementation details in the final version. For completeness:
>
> *Step size:* 3 mm (TractInferno), 1 mm (ISMRM), matching each dataset’s native resolution.
> *Cube size:* 3×3×3 CNN kernel to balance local context and memory constraints.
> *SH order:* 12, supports a wide range of acquisition schemes (gradient directions vary from 22 to 132 before resampling).
> *Data split:* ISMRM: streamlines are splitted 80% train, 20% val; TractInferno: 10 train subjects, 2 validation subjects, 1 test subject (limited by GPU RAM).
> *Streamlines:* ~200K seeds (ISMRM), ~1M seeds (TractInferno).
> *Preprocessing:* DWI resampled to 100 directions via SH order 12.
> *MRI resolution:* Native resolution used throughout.
> *Augmentation:* Streamline reversal applied during training.
>
> **4. Simple CNN architecture and lack of spherical-harmonic (SH) input ablation**
>
> We found that a single 3D convolutional layer effectively captures spatial context with minimal compute, as shown in Table 2 of the paper. Deeper CNNs may yield improvements but are beyond the scope of this work given memory and efficiency constraints. This will be discussed in the final manuscript.
> We agree that using SH coefficients directly is worth studying. We now include an ablation where SH coefficients (order 12) are used as input. See Table 4 in “Additional Experiments” below.
>
> **5. Dependence on teacher streamlines**
>
> This is an inherent limitation in all supervised methods relying on a single teacher. In our case, TractInferno reference bundles are generated by ensembling results from four different tracking algorithms, followed by RecoBundlesX extraction and ~200 hours of manual QA. This process reduces algorithmic bias and yields more reliable ground truth.
>
> **Response to Specific Questions**
>
> **1. Reference streamlines generation**
>
> - **ISMRM:** Manually segmented from an HCP subject, refined with anatomical masks, deterministic tracking, and post-processing. These are used *only* during training, never at test time. Furthermore, test-time trajectories differ due to random seeds, preventing memorization.
> - **TractInferno:** Ensembled outputs of four classical algorithms, filtered via RecoBundlesX, and curated with ~200 hours of manual QA.
>
> **2. 3×3×3 kernel choice**
>
> This kernel allows each voxel’s prediction to include 26 neighboring voxels, capturing critical fiber information (e.g., crossings, partial volumes). Larger kernels or deeper CNNs would increase memory costs and are currently infeasible under our hardware limitations.
>
> **3. Can the CNN be applied once to the full volume?**
>
> During training, the CNN weights are continuously updated, requiring feature recomputation at each step. During inference, our ablation shows that removing the CNN layer yields only a minor speedup, as the main bottleneck is the transformer's autoregressive nature. Moreover, extracting features for the entire volume would inefficiently allocate resources to non-white matter regions.
>
> **4. How is interpolation handled at off-grid points?**
>
> During training, all streamline points are mapped to integer voxel coordinates. During inference, off-grid predictions result in streamline termination. No interpolation is performed.
>
> **5. Fixed fODF or a path-dependent distribution prediction?**
>
> It predicts a path and context-dependent orientation distribution. The same voxel can yield different predictions depending on the streamline's history. We will clarify this point in the final manuscript.
>
> **6. Learned stopping criteria**
>
> Only the End-of-Fiber (EoF) token is learned. Others—angular threshold, FA, and anatomical constraints—are rule-based. Percentages for stopping:
>
> - Angular threshold: 70%
> - EoF: 65%
> - FA threshold: 10%
> - Outside image / WM mask / max length: 0%
>
> **7. Using SH-coefficients directly**
>
> We now evaluated this variant. See Table 4 and discussion below.
>
> **8. Can you identify where classical methods fail vs. your model, and vice versa?**
>
> Yes. As shown in our MITK comparison (Table 2 and 3 above), classical methods often fail in regions with complex fiber geometries (e.g., crossings, fanning), where our model excels. In contrast, our model can sometimes overreach in simple regions, reflecting a precision–recall trade-off.
>
> **9. Can the Tractometer metric be used on TractInferno?**
>
> In theory, yes, but it requires infrastructure adaptation. The ISMRM phantom remains a widely used benchmark for standardized, fair comparison across studies.
>
> ---
>
> **Additional Experiments: SH Coefficient Input Ablation**
>
> We retrained our model with SH coefficients inputs (order 12) instead of raw DWI. All other parameters remained the same.
>
> **Table 4: SH vs. Raw DWI**
> | **Model**                  | **VC (%)** | **OL (%)** | **OR (%)** | **F1 (%)**|
> |----------------------------|------------|------------|------------|------------|
> | Raw DWI (original)         | 81.51      | **83.72**| 30.83| 74.78|
> | SH Coefficients (order 12) | **81.98**  | 80.58| **24.28**  | **77.78**|
>
> SH input reduced overreach and improved F1 by +3%, with minor overlap drop. These results will be included and discussed in the final manuscript. We thank the reviewer for this suggestion.
>
> We appreciate the reviewer’s constructive comments and suggestions, which have helped us improve both the clarity and scope of our work. Thank you again for your thoughtful review.

---

> > ### Comment · Reviewer_Nf5j · 2025-08-04
> > **Thank you**
> >
> > Thank you for taking the time to address most of my concerns and questions. The rebuttal is strong, providing a deeper analysis of the strengths and weaknesses of the method, as well as useful practical clarifications. I am raising my score accordingly. However, the current draft of the paper requires major revisions to incorporate these new analyses, which would necessitate another round of review.

---

> > > ### Author Response · Authors · 2025-08-05
> > > **Acknowledging Revisions and Improvements**
> > >
> > > We sincerely thank the reviewer for their thoughtful follow-up and for acknowledging the strength of our rebuttal. We also appreciate the increased score and the recognition of the improvements we introduced.
> > >
> > > We agree that incorporating the new analyses and practical clarifications into the manuscript will substantially enhance its quality and clarity. We are committed to making these revisions thoroughly and transparently.
> > >
> > > The revised version will include all additional results and discussions, as well as implementation details and code for our inference-time optimization. Notably, we reduced inference time from **7 hours to just 43 minutes**, as detailed in our response to Reviewer **EBQD**. We believe this improvement significantly strengthens the practical utility of our method.
> > >
> > > We thank the reviewer again for their constructive and encouraging feedback.

---

### Official Review · Reviewer_XRRU · 2025-07-03

**Clarity:** 4
**Significance:** 4
**Originality:** 4
**Rating:** 5
**Confidence:** 4

**Summary:**

This paper describes an estimator of white matter tract structure
that uses a CNN to characterize diffusion MRI info locally,
along with a transformer that marches down the streamlines,
paying attention to what has come before.

Tractography is an important capability that enables research
in neuroscience.  Empirically we have seen that subtle abnormalities
in white matter structure are associated with important brain disorders.
Tractography is estimated from specialized MRI scans (MRI is non-invasive).

The authors summarize the background on what tractography is
and what characterizes the state of the art in the field.
Tractography, the process of estimating white matter structure
by constructing streamlines is very challenging, as the voxels
are quite large compared to the anatomical structure.  While there
have been many approaches to this problem, including, e.g., unscented
Kalman filters, the problem is not solved, witness the increasing
scores in the challenges.

The present contribution achieves excellent results by harnessing
the strengths of CNNs to summarize the diffusion image data locally,
and by using a transformer to model the sequential nature of
tractography (the classical method of tractography is sequential:
find the direction with the strongest evidence, take a step in
that direction, repeat).  This sequential nature suggests use
of a recurrent mechanism, and currently, transformers are the
most successful recurrent mechanisms.  For the local 'image'
analysis, CNNs are arguably the most successful mechanism.

The present paper matches the structure of the problem to two of
our most powerful mechanisms, transformers and CNNs, and does
so in an elegant and novel way that yields SOA results.

**Questions:**

One item I didn't quite get is:   tracking is started from random
seeds in the WM.   Are these seeds in the 'interior' of the WM
as a 3d structure?  If so, is tracking carried out in two directions,
one opposite the other?

In the description of the data, I was wondering how the ground truth
streamlines were generated?   Manual tracing by people?

**Ethical Concerns:**

["NO or VERY MINOR ethics concerns only"]

**Final Justification:**

I feel there is substantial novelty in bringing this hybrid method to
a problem of substantial import in neuroscience; that, and the
results argue for acceptance, for me.

**Limitations:**

yes

**Paper Formatting Concerns:**

None.

**Quality:**

4

**Strengths And Weaknesses:**

This is a very interesting and appropriate model for the problem at
hand.  It produces SOA results, and open source infrastructure will be
provided.  The paper is structured very well.  It clearly situates the
work in relation to what has come before.  The writing is
exceptionally clear.

---

> ### Author Rebuttal · Authors · 2025-07-29
>
> **Response to Reviewer XRRU**
>
> We sincerely thank the reviewer for the thorough assessment and for the exceptionally positive feedback regarding our methodology, results, and presentation. We appreciate your thoughtful questions and address them below.
>
> **1. Seeding and Tracking Directions:**
> Tracking is initiated from random seed points located within both the interior and, in some cases, the boundaries of the white matter. For each seed, tractography proceeds in a single direction, determined by the probability distribution predicted by our model based on local DWI measurements.
>
> **2. Ground Truth Streamline Generation:**
> *ISMRM 2015 Phantom Reference:* Ground truth bundles were created by manual segmentation from a Human Connectome Project subject’s tractogram, guided by expert anatomical definitions. Bundle-specific masks enabled deterministic tracking, with post-processing to remove implausible streamlines. Details are provided in the official Tractometer documentation.
>
> *TractInferno Dataset:* Reference bundles were generated using an ensemble of four tractography methods, followed by bundle extraction with RecoBundlesX and extensive manual quality control. About 200 hours of expert QC ensured anatomical plausibility and completeness, as detailed in the TractInferno paper and dataset documentation.
>
> We hope these clarifications are helpful. We appreciate the reviewer’s constructive comments and suggestions, which have helped us improve the clarity of our work. Thank you for your thoughtful review and support.

---

> > ### Comment · Reviewer_XRRU · 2025-08-04
> > **enthusiasm slightly moderated**
> >
> > I didn't notice the potential for leakage , though I applaud the additional evaluations.

---

> > > ### Author Response · Authors · 2025-08-05
> > > **On Evaluation Protocol and Generalization**
> > >
> > > We thank the reviewer for their thoughtful follow-up and for acknowledging the value of the additional evaluations.
> > >
> > > As discussed in our reply to Reviewers **Nf5j** and **EBQD**, we recognize that the **ISMRM 2015 evaluation involves training and testing on the same diffusion scan**. Although this setup has inherent limitations, it remains a **widely accepted protocol for ensuring comparability across methods**.
> > >
> > > That said, our model **does not rely on voxel-level memorization**: its predictions incorporate both local image context and the trajectory of the streamline up to the current point. Since test streamlines follow **different paths due to randomized seeding**, the model encounters novel input configurations even in familiar regions.
> > >
> > > To directly assess generalization, we further evaluated our method on **four held-out TractInferno subjects**. Despite being trained on just **10 subjects (vs. 198 for the baselines)**, our model achieved the **top Dice score on three subjects and second-best on the fourth**, demonstrating strong performance under limited training data (see Table 1 in our response to reviewer Nf5j).
> > >
> > > We appreciate the reviewer’s careful reading and constructive feedback.

---

### Official Review · Reviewer_EBQD · 2025-07-03

**Clarity:** 4
**Significance:** 3
**Originality:** 2
**Rating:** 4
**Confidence:** 4

**Summary:**

This study proposes a CNN-aided transformer-based approach to predict white matter fiber streamlines. The study intuitively poses tractography as a pathfinding algorithm, i.e., it predicts neural fibers from data and deals with challenges such as crossing and merging of fibre bundles. This leads to the formulation of the spatio-sequential formulation of the fibre orientation distribution function (fODF), where local features are captured by convolutional filters, which act as context for the transformer to make a prediction of the fODF in the streamlines. It does it in an autoregressive manner – i.e., it takes the voxel activity corresponding to the previous coordinates and makes the prediction about the next coordinate using a decoder-based transformer. For this, it minimises the KL-divergence between the discretized local streamline vector (soft-binned) and the predicted distribution over directions. This model is then probed like an autoregressive language model to generate the trajectory.

**Questions:**

* In the inference phase, in spite of taking the argmax over the predicted probabilities, the authors can see if the performance changes if we compute the expected direction. This converts discrete outputs to a continuous prediction space.
* What is the train-test split in the ISMRM dataset, as there appears to be only 1 scan? Or is the scan just a test sample.
* I am open to upgrading my rating if the limitations are addressed satisfactorily

**Ethical Concerns:**

["NO or VERY MINOR ethics concerns only"]

**Final Justification:**

I have read the authors' rebuttal responses and discussed these responses, am happy to upgrade my score.

**Quality:**

3

**Strengths And Weaknesses:**

Strengths:
* The literature review is well structured and contains important details of recent ML-based tractography methods.
* The approach used to extract local dwi features and use them for decoding using transformer decoders is intuitive and a plausible approach that could extract robust streamlines.
* The numbers show significant improvement over existing methods. Ablation studies were useful too, to see the significance of the parts of the model. Could become an important neuroscience tool.

Major Weaknesses
* The idea of using CNNs and Transformers for tractography prediction is not completely novel, as the authors themselves point out (Hosseini, 2022). Nonetheless, extending it to an auto-regressive model is the novelty (although that too is obtained by using traditional masking as done in LLMs). It would be better if the authors mentioned that more explicitly, just to avoid overselling the method. Although it can have a significant impact on tractography in neuroscience, the novelty in ML models/approaches is limited.
* The inference time for tractography is 7 hours for V100 GPUs, which is comparable to or longer compared to traditional tractography pipelines. Could the authors confirm that this is correct? This makes it difficult to justify as a useful method.
* All the results are on a single phantom scan. For human DMRI data, only 10 participants are used, with only 1 test subject. This makes it challenging to evaluate the usefulness of the algorithm at scale, in big datasets with thousands of scans.

Minor Weakness
* In equation 1, why is X(s_i) being used to make a prediction of the coordinate s_i. We need to predict the next coordinate based on the history of the coordinate. Should the output not be P(f|X(s1)… X(si-1))?
* Can the authors refer to the experiment section in the methods section for the details of the architecture? Incorporating them into a more detailed figure would be useful for the readers.

---

> ### Author Rebuttal · Authors · 2025-07-29
>
> **Response to Reviewer EBQD**
>
> We thank the reviewer for the detailed and constructive feedback. We appreciate your acknowledgment of the strengths of our work and your openness to revising the score if the limitations are properly addressed. We hope that our responses, clarifications, and new analyses below will fully address your concerns.
>
> **Addressing Main Weaknesses**
>
> **1. Novelty**
> We thank the reviewer for the feedback. We acknowledge that our method builds on established ML architectures, and we will revise the manuscript to better articulate how our adaptation to the tractography setting constitutes a meaningful and practical contribution.  While CNN and Transformer architectures have been explored in diffusion MRI tractography (e.g., Hosseini et al., 2022), our core novelty lies in adapting and integrating these tools into a sequential, context-aware, auto-regressive framework inspired by language models. Specifically, we reformulate streamline tracking as a sequence modeling task, treating each streamline as a “sentence” of diffusion-derived “tokens,” and predicting the next direction using causal self-attention over the streamline’s history. This enables the model to capture long-range dependencies and better handle complex fiber configurations such as crossings and fanning.
>
> To underscore the value of our contribution, we conducted an additional experiment comparing our method to the deterministic MITK Diffusion pipeline, which tracks streamlines based solely on local maxima without contextual history. Using the ISMRM 2015 dataset and its evaluation metrics, we focused on the Right Brainstem Pontine and Left Cingulum bundles, chosen for their anatomically complex structure with prominent crossings and branchings. As shown in Table 1, our model outperforms MITK in valid connections, overlap, and F1 score. A whole-brain comparison in Table 2 further demonstrates the benefit of our context-aware formulation, achieving significantly better results. We will include these results in the revised manuscript, along with visualizations of the reconstructed bundles.
>
> **Table 1: MITK vs TractoTransformer on challenging bundles (ISMRM evaluation tool)**
> *VC = valid connections, OL = overlap, OR = overreach.*
>
> | **Bundle**                    | **Method**        | **VC** | **OL (%)** | **OR (%)** | **F1 (%)** |
> |-------------------------------|-------------------|--------|------------|------------|------------|
> | Right Brainstem Pontine Tract | MITK              | 11103  | 39.48      | **18.37**  | 53.22      |
> |                               | TractoTransformer | **31996** | **88.31** | 27.36      | **79.71**  |
> | Left Cingulum Bundle          | MITK              | 7619   | 46.97      | **31.71**  | 55.66      |
> |                               | TractoTransformer | **16792** | **89.62** | 39.09      | **72.52**  |
>
> **Table 2: Whole-brain tractography comparison**
>
> | **Method**         | **VC (%)** | **OL (%)** | **OR (%)** | **F1 (%)** |
> |--------------------|------------|------------|------------|------------|
> | MITK               | 55.74      | 48.03      | **28.56**  | 54.98      |
> | TractoTransformer  | **81.51**  | **83.72**  | 30.83      | **74.78**  |
>
> **2. Runtime and Practicality**
> We acknowledge that the current end-to-end runtime (~7 hours on a single V100 GPU) presents scalability challenges. Our current implementation is research-oriented, prioritizing anatomical fidelity and accuracy over computational efficiency. That said, there is substantial room for optimization. Since streamlines are generated independently, inference time can be significantly reduced through straightforward parallelization across GPUs or CPU threads, enabling near-linear speedup. We are also actively developing strategies to further accelerate inference, including key-value caching for the autoregressive process and optimized batch processing. Alongside open-sourcing our code and model, we plan to release an optimized implementation incorporating these improvements to support more practical runtimes and wider adoption. We will revise the manuscript to better reflect these limitations and our plans to address them.
>
> **3. Evaluation Scale**
> ISMRM has long served as a widely used benchmark for tractography, owing to its high anatomical fidelity and controlled, reproducible setup, despite its limitation to a single phantom scan. To complement this, we initially included a qualitative evaluation on an unseen in-vivo scan from the TractInferno dataset, which provides a valuable alternative perspective. We agree, however, that broader quantitative evaluation across diverse datasets is important for assessing generalizability.
>
> To address this, we extended our TractInferno experiment to include quantitative evaluation on four test subjects. With the courtesy of the Tractinferno authors, we obtained per-subject evaluation scores (Dice, Overlap, and Overreach) for five baseline methods: Det-Cosine, Det-SE (Learn2Track), Prob-Sphere (DeepTract), Prob-Gaussian, and Prob-Mixture. These scores were not included in the original publication and were shared with us upon request (see the TractInferno paper for method details). Notably, our model was trained on only 10 subjects, whereas the TractInferno baselines were trained on 198. Despite this, as shown in Table 3, our method achieves the highest Dice score on three out of four subjects and second-best on the fourth, demonstrating strong generalization from limited data. Scaling to the full TractInferno test set was not feasible at the time of submission due to hardware constraints. Nonetheless, we will release our full codebase to support future benchmarking and reproducibility at scale.
>
> **Table 3: TractInferno test subjects — Dice, Overlap, Overreach**
>
> | **Subject** | **Metric**   | **Det-Cosine** | **Det-SE** | **Prob-Sphere** | **Prob-Gaussian** | **Prob-Mixture** | **TractoTransformer** |
> |-------------|--------------|----------------|------------|-----------------|-------------------|------------------|------------------------|
> | sub-1006    | dice         | 0.618          | 0.544      | 0.570           | 0.597             | 0.399            | **0.626**              |
> |             | overlap      | 0.575          | 0.470      | 0.525           | **0.593**         | 0.284            | 0.590                  |
> |             | overreach    | 0.281          | 0.231      | 0.248           | 0.381             | **0.074**        | 0.262                  |
> | sub-1019    | dice         | 0.614          | 0.558      | 0.597           | 0.606             | 0.381            | **0.654**              |
> |             | overlap      | 0.551          | 0.493      | 0.540           | 0.566             | 0.266            | **0.625**              |
> |             | overreach    | 0.239          | 0.198      | 0.227           | 0.266             | **0.051**        | 0.279                  |
> | sub-1024    | dice         | 0.585          | 0.593      | 0.600           | **0.624**         | 0.354            | 0.602                  |
> |             | overlap      | 0.511          | 0.493      | 0.533           | **0.579**         | 0.302            | 0.562                  |
> |             | overreach    | 0.196          | 0.153      | 0.192           | 0.245             | **0.039**        | 0.267                  |
> | sub-1061    | dice         | 0.618          | 0.606      | 0.616           | 0.620             | 0.430            | **0.629**              |
> |             | overlap      | 0.560          | 0.523      | 0.562           | 0.575             | 0.314            | **0.580**              |
> |             | overreach    | 0.256          | 0.180      | 0.209           | 0.259             | **0.068**        | 0.244                  |
>
> **4. Equation Clarification**
> We thank the reviewer for bringing this to our attention. At each $s_i$, we predict the fODF conditioned on entire preceding path up to $s_i$, and then advance to $s_{i+1}$ in the selected direction. We will correct the indexing and clarify this in the final manuscript.
>
> **5. Manuscript Navigation and Figures**
> We agree that clearer section references and a more detailed architecture diagram will improve readability. We will revise the Methods section and figures accordingly.
>
> **Response to Specific Questions**
>
> **ISMRM Split:**
> Reference streamlines were randomly split 80/20 into training and validation. These were used for supervision only, not during inference. While voxel positions may recur, the model conditions on full streamline context, making it highly unlikely to encounter identical input-output pairs. Generalization is further supported by results on unseen subjects (both qualitative and quantitative).
>
> **Expected Direction Inference:**
> Thank you for this suggestion. We evaluated tracking using the expected direction under the predicted fODF, instead of argmax, using identical seeds and evaluation. Results are similar: argmax improves F1 and overlap, while expected direction reduces overreach.
>
> **Table 4: Mode vs Expected Direction on ISMRM**
>
> | **Model**            | **VC (%)** | **OL (%)** | **OR (%)** | **F1 (%)** |
> |----------------------|------------|------------|------------|------------|
> | TractoTransformer    | 81.51      | 83.72      | 30.83      | 74.78      |
> | Expected direction   | 79.37      | 75.43      | **25.52**  | 73.44      |
>
> We thank the reviewer once again and will integrate this comparison and all clarifications, revisions, and new results into the updated manuscript. We appreciate the reviewer’s constructive comments and suggestions, which have helped us improve both the clarity and scope of our work. Thank you for your thoughtful review.

---

> ### Comment · Reviewer_EBQD · 2025-08-04
> **Rebuttal clarifies some questions, but raises others**
>
> I thank the authors for their earnest efforts with answering my questions. Yet, I am not strongly convinced that the algorithm reflects a scalable and important use case for the community. My remaining concerns are as follows:
>
> i) **Novelty by itself does not suffice**: While modeling streamline tracking as sequence generation appears to reflect a novel approach, such novelty has to be backed up by substantial advantages in terms of performance -- either accuracy or runtime. Otherwise, a novel conceptual idea cannot by itself be a selling point.
>
> ii) **Concern about accuracy**: The new evaluation numbers suggest that the authors' method suffers from substantially greater over-reach than competing methods -- this is clear both with the ISMRM and the TractInferno results. As I'm sure the authors are aware, (higher) overlap must be balanced with (lower) overreach. The Dice/F1 scores ameliorate these concerns only partially, at best.
>
> iii) **Runtime remains prohibitively slow**: Although I appreciate the authors addressing the question about runtime, the claim that "inference time can be significantly reduced through straightforward parallelization across GPUs" is misleading at best. Developing such runtime improvements can be prohibitively challenging. These require specialized and careful analyses of the underlying operations while respecting physical constraints (crossing/duplicate fibers), often followed by innovative steps for hardware translation (see e.g. Caiafa and Pestilli, Scientific Reports, 2017).
>
> Without a clear demonstration of significant advantages, the very high runtimes precludes the potential utility and adoption of this method for tractography, even for research purposes in the lab, where 100s, if not 1000s of connectomes must be generated for each study.

---

> > ### Author Response · Authors · 2025-08-05
> > **Addressing Remaining Concerns**
> >
> > We thank the reviewer for their time and valuable feedback. Below, we address the remaining concerns point by point.
> >
> > **i) Runtime and Scalability:**
> > As previously described, our model generates streamlines auto regressively using a decoder-only Transformer. Each prediction step depends solely on the streamline’s prior trajectory and is independent of other streamlines. This design allows for straightforward parallelization across seed points and further optimization via key-value (KV) caching, a well-established technique for accelerating inference in decoder-only Transformer architectures.
> >
> > To directly address the reviewer’s concern, we implemented and integrated both techniques:
> >
> > - **Multi-GPU inference:** We parallelized inference across **four V100 GPUs** by partitioning seed points across independent processes. Each process generates streamlines in isolation, after which results are merged into a single tractogram.
> >
> > - **Key-value caching:** We added a **KV-caching mechanism** to our trained Transformer model, avoiding redundant attention computations during inference. **No model retraining is required**, and this yields **identical output tractograms**.
> >
> > These optimizations yielded a **~10× speedup, reducing whole-brain tractography time from 7 hours to just 43 minutes**. Specifically, runtime improved **nearly linearly with GPU count (~4× on 4 GPUs)**, with an additional **~2.5× speedup from KV caching**.
> >
> > These improvements were made post-submission and will be included in our final manuscript and open-source code release. We believe this demonstrates that our model is **practically scalable and well-suited for large-scale applications**. Further speedups are possible via batched GPU inference, higher-memory hardware, or advanced inference techniques such as speculative decoding, which we plan to explore in future work.
> >
> > **ii) Accuracy and Overreach:**
> > We agree that overlap and overreach capture a fundamental tradeoff in tractography: achieving good coverage of true connections while avoiding false positives. The **Dice score reflects this tradeoff**, balancing both overlap and overreach to provide an overall assessment of quality.
> >
> > - On **TractInferno**, the method with the best overreach (**Prob-Mixture**) obtained the **lowest Dice scores**, performing **tens of percentage points below ours**. The second-best overreach method (**Det-SE**) also achieved **considerably lower Dice scores**. While our overreach is higher than these two methods, it remains comparable to the remaining three. **Importantly, our method achieves the highest Dice score on 3 of 4 subjects**, indicating strong performance in both precision and coverage.
> >
> > - On the **ISMRM 2015 challenge**, our method achieves equal or improved overreach relative to prior approaches. Specifically, our overreach matches Random Forest (RF) and surpasses both Learn2Track and Entrack. Please refer to **Table 1 of the main paper** for details.
> >
> > **iii) Novelty:**
> > We agree that our conceptual contributions must translate into tangible performance benefits, whether in accuracy, runtime, or both.
> >
> > - **Accuracy:** Accurately resolving fiber crossings and branchings is a key challenge in tractography. Our sequential, context-aware autoregressive framework specifically addresses these complex fiber configurations. As shown in **Table 1 and Table 2 of the rebuttal**, our model outperforms classical methods that rely on static FODFs and lack contextual modeling. It also achieves competitive or improved results compared to recent state-of-the-art approaches on both ISMRM and TractInferno benchmarks.
> >
> > - **Runtime:** While autoregressive models are often associated with high computational cost, we demonstrate that our approach remains efficient. As described above, our optimizations resulted in a **~10× speedup, reducing whole-brain tractography time from 7 hours to just 43 minutes**.
> >
> > Thank you again for your thoughtful and constructive review. We would be happy to receive any further feedback to help strengthen the final version of our work.

---

> > > ### Comment · Reviewer_EBQD · 2025-08-06
> > >
> > > Thank you for engaging in a conversation, and seeking to address these questions. My suggestions are below:
> > >
> > > > We parallelized inference across four V100 GPUs by partitioning seed points across independent processes. Each process generates streamlines in isolation, after which results are **merged into a single tractogram** (emph. added).
> > >
> > > i) **Parallelizing runtime**: I apologize if I did not phrase my concern clearly enough. Certainly *naive parallelization* can be done for any method (e.g. MRTrix) -- for example, by seeding different ROIs across different servers even, which could bring the time down to seconds. But this would lead to the same issues that I mentioned in my earlier comment (#iii) -- duplicate fibers and uncontrolled fiber crossings. This happens because the processes on the different servers aren't talking to each other and are generating tractograms independently. After this, a key challenge happens with the highlighted step above -- "merg[ing] into a single tractogram".
> > >
> > > The reference that I cited previously (Caiafa et al, 2017/LiFE), and other related methods like SIFT, deal precisely with this challenge of *post hoc* pruning of an overabundance of streamlines to create a single tractogram, maintaining high overlap, without high overreach. Coming up with a way to do this online -- during the tractography generation -- requires non-trivial computations and logic. Even when the seeded regions are naively parallelized they must communicate with each other to avoid redundant streamlines, and other artifacts, and the time needed for such communication overheads must be measured.
> > >
> > > For the purposes of this revision, I would be happy if the authors could report comparisons with standard tractography pipelines like Mrtrix and show how fast their method is -- on the same hardware (type, # GPUs) for fair comparisons.
> > >
> > > ii) **Accuracy/Dice score**: I agree that the Dice score seeks to address the tradeoff between overlap and overreach, but like I mentioned in my previous comment, it does so "only partially". To be more specific, recent studies have shown that the Dice score is often biased by how conservative the baseline method is. For example, methods that generate a low proportion of false positives (conservative methods) tend to have lower Dice scores and vice versa. See for example: Raina et al (2023), Tackling Bias in the Dice Similarity Coefficient; Maier-Hein et al (2022). Metrics reloaded: Pitfalls and recommendations for image analysis validation. In other words, methods that err on the side of generating more false positives (more overlap at the cost of overreach) could have inflated Dice scores.
> > >
> > > An ideal way to show superior performance would be to plot overlap versus overreach for different competing methods, examine the Pareto front, and show that the proposed method falls ahead of this pareto front.

---

> > > > ### Author Response · Authors · 2025-08-07
> > > > **Parallel Tractography and Pareto Front Evaluation**
> > > >
> > > > Thank you for your thoughtful feedback and suggestions.
> > > >
> > > > **1. Runtime**
> > > > We understand and appreciate your concerns regarding the challenges of parallelization in classical tractography methods, where independent processes can lead to issues such as redundant streamlines or uncontrolled crossings.
> > > > However, in **our method, streamlines are seeded and advanced completely independently, with no inter-process dependency at any stage**. Importantly, **the exact same tractogram we originally reported as taking 7 hours to generate is reproduced identically in just 43 minutes** after straightforward parallelization and optimization steps. Thus, increased overreach or other artifacts raised in your comments cannot be attributed to parallelization, as there is **absolutely no change to the tractogram content**.
> > > >
> > > > Regarding comparisons with standard tractography pipelines, we fully agree that benchmarking against widely used tools like MRtrix on the same hardware would be informative. However, setting up MRtrix for GPU-based tractography with identical preprocessing and evaluation pipelines would require substantial engineering effort, which is unfortunately beyond the scope of this revision cycle.
> > > > That said, we can report that the **MITK tool** used in our quantitative evaluation produces tractograms in approximately **8 minutes on CPU**, but, as shown in **Tables 1 and 2 of our rebuttal, yields substantially lower anatomical accuracy** compared to our method. As this work focuses on improving the biological plausibility and fidelity of tractography, **our current emphasis remains on anatomical accuracy rather than runtime optimization**.
> > > >
> > > > **2. Accuracy**
> > > > We appreciate the reviewer’s thoughtful comment regarding the limitations of the Dice score. To better capture the trade-off between Overlap (OL) and Overreach (OR), we followed the reviewer’s suggestion and **plotted the average OL-OR values across all test subjects for the five competing methods**. We then **computed the empirical Pareto front from these baselines**. **Our method lies above this front, achieving an Overlap gain of 0.0241 over the Pareto-optimal baseline at the same OR, demonstrating a more favorable balance between sensitivity and specificity than Dice alone conveys**. We also note that these results were obtained using a model trained on only 10 subjects, compared to the 198 subjects used to train the competing methods.

---

> > > > > ### Comment · Reviewer_EBQD · 2025-08-07
> > > > >
> > > > > Thank you for the clarifications.

---

### Note · Authors · 2025-08-13

**Author Final Remarks**

We thank the reviewers and AC for their constructive feedback and for the productive discussion phase. Below we summarize the key additions, clarifications, and new results introduced since submission.

**1. Generalization on Additional Subjects** – Beyond the original evaluation, we extended our TractInferno experiments to four held-out test subjects, using per-subject metrics provided by the dataset authors for five leading baselines. Despite training on only 10 subjects (vs. 198 for baselines), our method achieved the highest Dice score on 3/4 subjects and second-highest on the fourth, confirming strong generalization from limited training data.

**2. Demonstrating contribution in terms of overlap vs. overreach** – As suggested by Reviewer EBQD, we plotted the Overlap vs. Overreach trade-off for all TractInferno baselines and computed the empirical Pareto front. Our method lies above this front, achieving an Overlap gain of +0.0241 at matched Overreach, indicating a more favorable sensitivity–specificity balance than competing methods.

**3. Demonstrating tracking results in complex fiber configurations** – We compared our method against the widely used MITK deterministic tractography pipeline on the ISMRM 2015 phantom. In anatomically complex bundles with prominent crossings (Right Brainstem Pontine, Left Cingulum), our model achieved substantially higher valid connections, overlap, and F1, demonstrating its advantage in resolving complex fiber geometries.

**4. Significant runtime improvement** – We implemented multi-GPU inference and key-value caching in the Transformer decoder, reducing whole-brain runtime from ~7 hours on a single V100 GPU to 43 minutes on 4×V100s, with identical tractograms. These improvements, to be released with our code, demonstrate that our method is practically scalable for large-scale studies.

**5. Spherical Harmonics Input Ablation** – Following Reviewer Nf5j’s suggestion, we evaluated a variant using spherical harmonic coefficients (order 12) directly as input. This reduced overreach and improved F1 by +3% compared to raw DWI input, confirming the value of this representation.

We will incorporate these results, clarifications, and limitations into the final manuscript, along with detailed figures, methodology updates, and optimized code release. We believe these additions strengthen the novelty, accuracy, and practical utility of our work, and address the main concerns raised during the review.

---

### Decision · Program_Chairs · 2025-09-17

**Decision:**

Accept (poster)

**Comment:**

This paper presents a method for estimating white matter tract structure by combining a convolutional neural network (CNN) to locally characterize diffusion MRI signals with a transformer that sequentially processes streamlines, attending to prior context along the path.

The approach is evaluated on two datasets and yields both quantitatively and qualitatively plausible tractograms. In particular, the qualitative results on the TractoInferno benchmark are very promising. Nonetheless, there are some concerns regarding the small sample sizes used during training, validation, and testing, as well as the computational cost of the method.

While the use of CNNs and transformers for tractography is not entirely new, the extension to an auto-regressive formulation adds a degree of novelty. The paper is well-structured and clearly written. I recommend it for publication.